# A Five-Year Malaria Prevalence/Frequency in Makenene in a Forest–Savannah Transition Ecozone of Central Cameroon: The Results of a Retrospective Study

**DOI:** 10.3390/tropicalmed9100231

**Published:** 2024-10-07

**Authors:** Joël Djoufounna, Roland Bamou, Juluis V. Foyet, Laura G. Ningahi, Marie P. A. Mayi, Christophe Antonio-Nkondjio, Timoléon Tchuinkam

**Affiliations:** 1Vector Borne Diseases Laboratory of the Research Unit for Biology and Applied Ecology (VBID-RUBAE), Department of Animal Biology, Faculty of Science of the University of Dschang, Dschang P.O Box 67, Cameroon; joeldjoufounna@gmail.com (J.D.); visnelfoyet@gmail.com (J.V.F.); lauritaningahi@gmail.com (L.G.N.); mayimariepaulaudrey@yahoo.com (M.P.A.M.); timotchuinkam@yahoo.fr (T.T.); 2Institut de Recherche de Yaoundé (IRY), Organisation de Coordination pour la lutte Contre les Endémies en Afrique Centrale (OCEAC), Yaoundé P.O. Box 288, Cameroon; antonio-nk@yahoo.fr; 3Laboratory of Malaria and Vector Research, National Institute of Allergy and Infectious Diseases, National Institutes of Health, 12735 Twinbrook Parkway, Rockville, MD 20852, USA; 4Department of Microbiology, University of Yaoundé I, Yaoundé P.O. Box 337, Cameroon

**Keywords:** malaria prevalence, Makenene, forest–savannah area, Central Cameroon

## Abstract

Objective: Understanding the epidemiological features of malaria is a key step to monitoring and quantifying the impact of the current control efforts to inform future ones. This study establishes the prevalence and frequency of malaria in a forest–savannah ecozone for 5 consecutive years in Cameroon. Methods: A retrospective study was conducted in 3 health centers of Makenene from 2016 to 2020, a period covering the second long-lasting insecticide net mass distribution campaign. Malaria infectious records were reviewed from laboratory registers. The difference in exposure to malaria was estimated using a regression logistic model. Results: A total of 13525 patients underwent malaria diagnostic tests, with a general malaria prevalence of 65.3%. A greater prevalence of malaria was observed in males (68.39%) compared to females (63.14%). The frequency of consultations in health centers was dominated by females, with a gender ratio (M/F) of 0.66. Annual trends in malaria prevalence slightly varied from 2016 to 2020, exceeding 60%: 65.2% in 2016; 66.7% in 2017, 68.1% in 2018, 63.2% in 2019, and 65.3% in 2020, with a significant seasonal variation (*p* < 0.0001). The highest malaria prevalence was observed during the short rainy season, no matter the year. Among positive cases, the most represented age groups were 6–15 (*p* < 0.0001), followed by those under 5, while the age group >25 years was the least represented. Conclusion: Close monitoring and additional intervention measures for malaria control are needed, as are more studies on vector bionomics and transmission patterns.

## 1. Introduction

Malaria remains a major public health problem, particularly in sub-Saharan Africa, which accounts for almost 96% of all worldwide cases [1]. Children under 5 and pregnant women remain the population groups with the highest risk of contracting the disease [1]. Disease burden can vary across short distances, between countries and neighboring villages or within a single settlement [2], and determines the intensity of malaria transmission, which is known to mirror the nature of the ecological zone [3]. In Cameroon, malaria is the first reason for hospital consultations, and the fight against the disease relies on the use of long-lasting, insecticide-treated nets (LLINs) and artemisinin-based combination therapy (ACT). The whole country is at risk, with different levels of endemicity across different ecological settings, ranging from hypo-endemic areas to hyper-endemic areas [4]. The country, from the southern part to the northern part, consist of: the equatorial forest with hyperendemic malaria [5], the savannah zone with hyperendemic malaria, and the northern Sahelian zone with hypo-endemic malaria [4]. Within these different zones, the epidemiological pattern of malaria transmission is influenced by the nature, and the type of biotope [6,7] by human activities [8,9] and forest–savannah intermediate localities, which is the case of Makenene [10]. Extensive farming practices, as well as environmental and climatic conditions, can have an impact on the epidemiology of malaria transmission [11]. In Cameroonian forest areas, the climate is favorable for the proliferation of anophelines vectors and perennial malaria transmission [4]. This is not the case in the savannah zone, where malaria transmission is seasonal and which, nevertheless, records more significant pathological consequences due to malaria than the forest. 

Considered a forest–savannah transition zone (from the forest massifs of the Mbam subdivision to the western grassland savannah), the locality of Makenene is mainly made up of endogenous people, with a large agricultural workforce. The city is a crossroads between different parts of the country where travelers (coming from high to low transmission settings and vice versa) usually stop over during the day and night to rest. Intense agricultural activities not only modify the environment, but can lead to increased resistance of malaria vectors to insecticides used in public health. A cross-sectional malaria study carried out in the year 2021 on asymptomatic people in this locality showed a high prevalence of asymptomatic Plasmodium falciparum malaria among both net-users and non-users [10], raising questions about the effectiveness of LLINs used in this locality. Indeed, the population has benefited from the 2011 and 2016 LLINs mass distribution campaigns. However, information on the evolution of malaria frequency from the beginning of the LLINs distribution campaign until the potential expiration date (which most often corresponds to renewing LLINs) is lacking. Therefore, this study aimed to determine the malaria trend and prevalence from 2016 to 2020 in Makenene health facilities. Such information is relevant to evaluating or assisting in efforts for the control and prevention of the disease.

## 2. Materials and Methods

### 2.1. Study Design

The study was conducted in the Makenene health area (Figure 1). This health area is located in the Mbam and Inoubou Division in the Central Region of Cameroon. The climate is equatorial and humid, and is divided according to the intensity of rainfall into two dry seasons and two rainy seasons of unequal length (with rainfall up to 721 mm/year): a long dry season (mid-October to March), a short dry season (June to mid-August), a long rainy season (mid-August to October), and a short rainy season (March to June) [12,13]. The highest rainfall is observed during the long rainy season, then in the short rainy season, and finally in the short and long dry seasons [12,13]. The health area of Makenene, according to the Ministry of Public Health (MoH), is made up of 3 health centers: The District Medical Center (DMC), the Baptist Church Health Center (UEBC health center or BCHC), and the Catholic Health Center (CHC) [12]. These health centers have been accessible to the population for more than 30 years. Based on its geographical location and the level of malaria endemicity in the central region of Cameroon, the health area of Makenene is situated in a forest–savannah transition zone, and is considered hyper-endemic [12]. The population who lives there consists of several ethnic groups, namely, the nationals of Kinding and Nyokon, who are indigenous, and then the Bamileke, Bamoun, Bantoum, and Anglophone, considered as non-native. This population is mostly made up of Christians and farmers, as described by [12,14]. 

### 2.2. Data Source

Malaria cases are both clinically and biologically diagnosed in Makenene health centers. All patients who have undergone a laboratory test are registered in a physical database that supports each health center (DMC, CHC and BCHC). This study included all malaria cases which were diagnosed using microscopy, RDT, or both from 2016 to 2020 in the Makenene health area. The data used are those transmitted from health centers to the Ministry of Public Health for statistics, confirming their reliability.

### 2.3. Data Collection 

A retrospective study on malaria cases from 2016 to 2020 was carried out in the 3 health facilities of the Makenene health area. Three collectors extracted data from the laboratory registers and registered on a previously established collection sheet. Variables recorded on the collection sheet included the health structure (District Medical Center, Baptist Church Health Center, and Catholic Health Center), the examination date, the gender and age of the patients, and the results of the malaria tests (Appendix A). 

### 2.4. Data Analysis

Data were entered into Excel 2016 spreadsheets, processed, coded, and imported into R software (R version 4.2.3, 10 January 2023) for analysis, following the flowchart of Figure 2. Patients with malaria-based diagnostic test results and variables such as age and gender were included in the analyses. Graphical analyses were performed to assess the trend of malaria prevalence per month for each year, as well as the trend of malaria prevalence per age group per month for each year, using the ggplot 2 package. Malaria prevalence was calculated by the ratio of positive cases out of the total number of patients per month during each year and per age group per month yearly. The difference in exposure to malaria was estimated through odds ratios and associated 95% confidence intervals from the logistic regression model. The analyses were considered significant at a significance level of 0.05.

## 3. Results

### 3.1. General Characteristics of Data Collected 

Data from 13,525 patients were collected from laboratory records from January 2016 to December 2020 in the Makenene health area. Very few patients were excluded due to missing data for their categorical variables (gender, age, and malaria status). In this dataset, 99.61% (n = 13,473) had all the results from the malaria diagnostic tests. Complete data on gender (99.26%, n = 13,426) showed that 99.17% (n = 13,414) had the results of the malaria diagnostic test. Of the 97.28% (n = 13,107) with data on age, 96.54% (n = 13,057) had the results of the malaria diagnostic tests, and 96.14% (n = 13,107) had data from the gender and malaria test results (Figure 2). The average ages of the patients varied between years and were 20.94 ± 20.50, 21.33 ± 20.54, 21.45 ± 20.17, 20.69 ± 19.94, and 21.49 ± 20.18 years in 2016, 2017, 2018, 2019, and 2020, respectively. The distribution rate of patients by gender was dominated by women in all health centers (Figure 1). Globally, 39.67% (n = 5326) were males and 60.33% (n = 8100) were females, with a gender ratio (M/F) of 0.66.

### 3.2. General Malaria Prevalence and Variation over Time (2016-2020) 

Annual trends in malaria prevalence slightly varied from 2016 to 2020 (Figure 3) (65.17% in 2016; 66.73% in 2017; 68.08% in 2018; 63,17% in 2019; 65.29% in 2020), with no significant difference observed (*χ*^2^= 12, *p* = 0.14). The monthly variation in the prevalence was noticed over the first years: The highest prevalence peaks were found during May and June (84.74% and 85.42%, respectively) in 2018 and in April 2016 (79.56%). The lowest prevalence peaks were found in August and October 2016 (43.09% and 43.75%, respectively). 

### 3.3. Seasonal Malaria Prevalence/Frequency and Variation over Time (2016-2020)

A seasonal variation in malaria prevalence was observed over the 5 examined years, with a significant difference (*p* < 0.0001) (Figure 4). The highest malaria prevalence was observed during the short rainy season, whatever the year (with the peak obtained during year 2018), while the lowest prevalence was observed during the long rainy season, whatever the year (Appendix A). 

### 3.4. Distribution of Malaria Prevalence by Gender and Age Group over Time (2016–2020)

The prevalence of malaria was grouped by age according to time and is presented in Figure 5. Irrespective of the year, malaria’s prevalence varied monthly within each age group. The highest prevalence was found in the age groups of 6–15 (80.44%) and 0–5 (74.61%), while the age group >25 years had the lowest prevalence (Appendix A). 

When taking into account both gender and age (Table 1), the highest prevalence rates were obtained for males in 2016 (81.82% of individuals in the 6–15 age group), in 2017 (86.13% of individuals in the 6–15 age group), and in 2019 (82.66% of individuals in the 6–15 age group), and among females in 2018 (82.37% of patients under 5 years old) and in 2020 (78.52% of patients in the 6–15 age group). 

### 3.5. Distribution of Malaria Prevalence by Health Center 

Table 2 presents the trend of malaria prevalence by health center (2016 to 2020). The highest malaria prevalence was obtained at the Baptist Church Health Center (BCHC) in 2018 (83.43%, with a positive case/negative case ratio of 5.04), then in 2017 (81.50% with a positive case/negative case ratio of 4.4), although this health center received fewer patients than the District Medical Center (DMC) and the Catholic Health Center (CHC). The lowest malaria prevalences (57.30% in 2016 and 58.34% in 2019), and, therefore, the lowest positive/negative malaria case ratios (1.34 in 2016 and 1.4 in 2019), were observed at the CHC. 

### 3.6. Determinants Associated with Malaria Prevalence

Globally, it appears from the results of the OR presented in Table 3 that, regardless of gender (male or female), the probability of testing positive for malaria was higher in younger groups (≤5 and 6 to 15) compared to the others (16 to 25 and >25). The age groups of 16–25 and >25 years were significantly less exposed to malaria compared to the age group of 0–5. Table 3 also shows that the age group of 6–15 was more likely to be exposed to malaria than the youngest age group in males (OR = 1.42 and *p* < 0.001) and in females (OR = 1.26 and *p* = 0.011).

Overall, a significant difference in the general malaria prevalence was observed between genders. Males were more likely to test positive for malaria than females (OR = 1.08; 95% CI = 1.02–1.14; *p*-value = 0.004). Moreover, a general prevalence of 68.39% (3639/5321) was observed in males, while a general prevalence of 63.14% (5110/8093) was observed in females (Table 4).

## 4. Discussion

To achieve the WHO objectives aiming at controlling malaria to limit the morbidity and mortality caused by the disease, it is important to know the epidemiological history and the trend of the disease promptly. Based on hospital databases and records, this study established malaria’s prevalence over 5 years in the Makenene health area. The results indicated that malaria was a major public health problem in the locality from 2016 to 2020. Over these 5 years, a hospital malaria prevalence of about 60% was reported during each year. Knowing that part of the population did not consult or visit the hospital in case of malaria, this prevalence is under-estimated for the entire community, or may represent only severe cases leading to consultation. 

The results show a high malaria prevalence of about 60% each year. This may be explained by the fact that only symptomatic cases were recorded during this study. However, even among asymptomatic people from the same locality, a prevalence of 38.41% was observed in 2021 [10]. The locality is crossed by an irrigation system called, in the local language, “Mock”, which could be responsible for the proliferation of mosquito breeding habitats, as seen elsewhere [11,15,16]. In addition, the locality is made up of poor-quality habitat construction. Doutum et al. [17] explained that malaria is a disease caused mainly by poverty, especially in rural and agricultural areas where poor-quality housing is observed, creating an environment conducive to the spread of malaria. Similar studies have shown a high prevalence (65.8%) in some health facilities in Uganda [18], and an average prevalence in southwest Ethiopia (33.4%) [16], in northwest Ethiopia (32.6%) [19], and in Central Gabon (33.5%) [20].

Malaria’s prevalence was slightly higher during the short rainy season than during the other seasons, regardless of the year. This could be explained by the fact that during this season, the pools of water (potential breeding sites) which favor the development of anophelines are more numerous, closer to homes, dry up less quickly than observed in the dry seasons, and are not disturbed by the excessive rainfall observed during the long rainy season, which sometimes washes out the larval habitats. Nevertheless, the prevalence observed during the other seasons is not negligible and could be attributed to the irrigation system made up of streams that surround the locality, which is perennial. These results are similar to those of some studies conducted on the impact of seasonality on malaria’s transmission and prevalence [16,20,21].

It is difficult to compare the results of malaria prevalence according to health centers because of their proximity. However, the result that the highest number of patients went to the DMC could be attributed to the fact that it is the reference health facility in the locality and the oldest. In addition, the prices of care are lower, which may attract more patients for consultations. In the event of an emergency, patients tend to go to the least populated health facility to be quickly treated, which could explain the higher prevalence observed at Baptist Church Health Center (BCHC) compared to other health centers.

The age group of 6-15 years was more likely to test positive for malaria, followed by people under 5 years in general. This is in agreement with studies conducted in the same locality, but on asymptomatic people [10]. The partially acquired immunity that begins to develop during childhood in such high-malaria-transmission areas might have a protective role in the age group above 15 years [22]. An association between age and malaria prevalence was also observed in previous studies [23,24]. Furthermore, a recent study on genotyping of anopheles’ mosquitoes blood meal showed that children under five transmitted more *P. falciparum* infections to mosquitoes, both relatively and absolutely [25].

Health facilities were mainly visited by female patients. Nevertheless, a higher malaria prevalence was observed in males. The life of the community in Makenene totally depends on farming, and most of the time, some young males spend their time (the whole evening) on the farm, while others are involved in early night activities like transport by motorbikes and, therefore, have a higher risk of exposure to anophelines’ malaria vector bites. Some authors have shown the ease women have compared to men in going to health facilities even when the symptoms are mild [18], which could also explain the prevalence observed in the age groups > 25 years. Similar results were observed in southwest Ethiopia [16]. It should be noted that this study was based on formal data and does not sufficiently allude to the fact that the majority of patients do not go to the health facilities where the registers are consulted. Data on LLINs’ population coverage, their use, and the effectiveness of these LLINs over time could be correlated with the prevalence/frequency of malaria observed during this period of 2016–2020. In addition, environmental parameters and even cases of recrudescence (following treatment or not) would have been considered to be able to justify the high malaria prevalence/frequencies in the locality. This study nevertheless shows that malaria remains a major public health problem in Cameroon in general, and in Makenene in particular.

## 5. Conclusions

A high malaria prevalence was noted from hospital databases in Makenene between 2016 and 2020. Although the population of this locality was covered by the second LLIN distribution campaign since 2016, there was residual malaria transmission capable of maintaining a high level of malaria endemicity in this locality. The most detrimental consequences were observed in children aged 6 to 15 years, then 0 to 5 years, where the highest frequency and prevalence were observed. Public authorities must combine other control methods with the current LLINs used to reduce this high malaria prevalence observed in Makenene. Additional studies on the bionomics of malaria vectors are also necessary to characterize malaria transmission and, thus, identify the mechanisms that sustain such a high prevalence of malaria in this locality.

## Figures and Tables

**Figure 1 tropicalmed-09-00231-f001:**
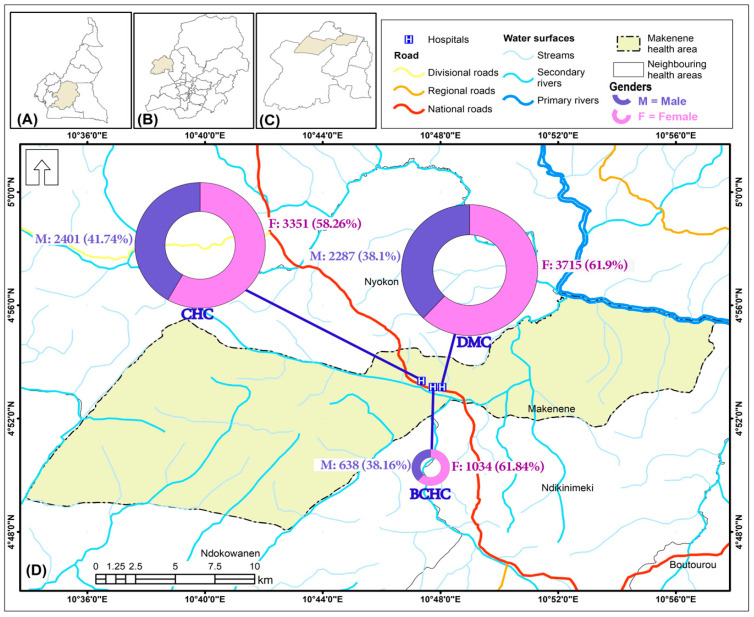
Map of the study site showing general malaria consultations by gender (2016–2020): (**A**) Cameroon, (**B**) Central Region; (**C**) Ndikinimeki health district; (**D**) Makenene health area.

**Figure 2 tropicalmed-09-00231-f002:**
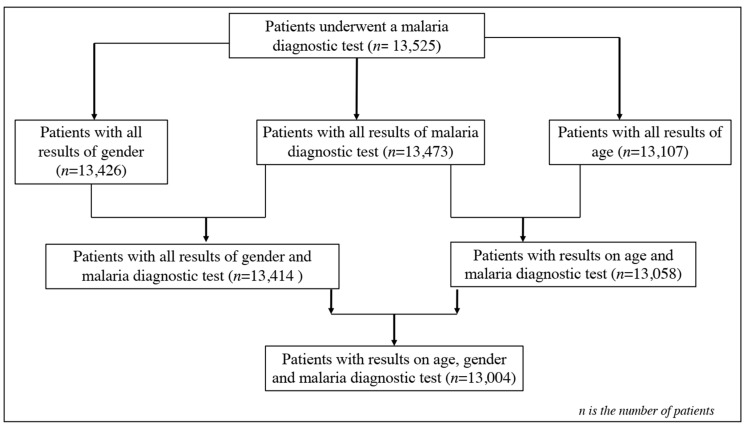
Flowchart of patients’ enrolment for data analysis.

**Figure 3 tropicalmed-09-00231-f003:**
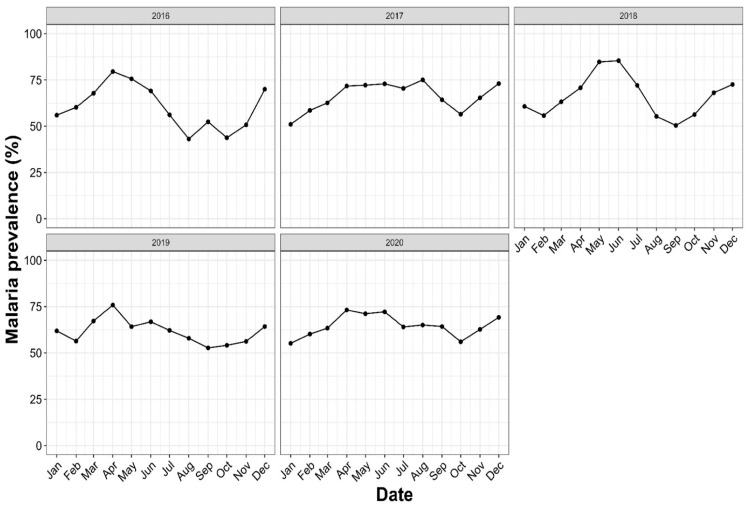
Annual trends of malaria prevalence in Makenene health services, Central Cameroon, from 2016–2020. The x-axis represents the date, expressed by the month, while the y-axis represents the malaria prevalence determined by the ratio of positive cases over the number of persons who visited the hospital with malaria symptoms during a specific period (year or month).

**Figure 4 tropicalmed-09-00231-f004:**
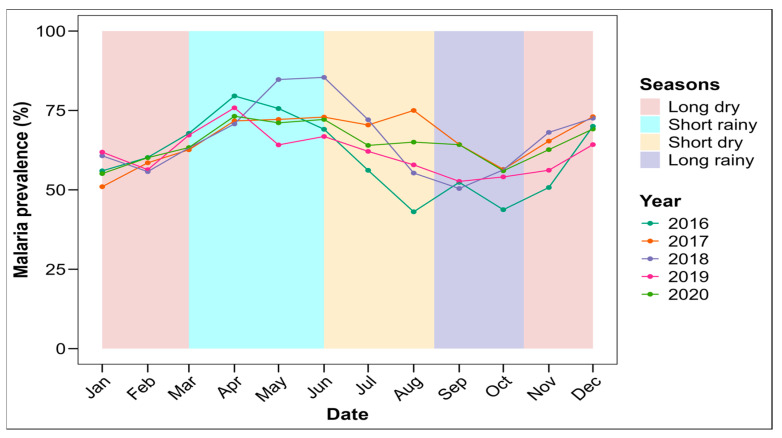
Seasonal trend of malaria prevalence in Makenene health area (2016–2020).

**Figure 5 tropicalmed-09-00231-f005:**
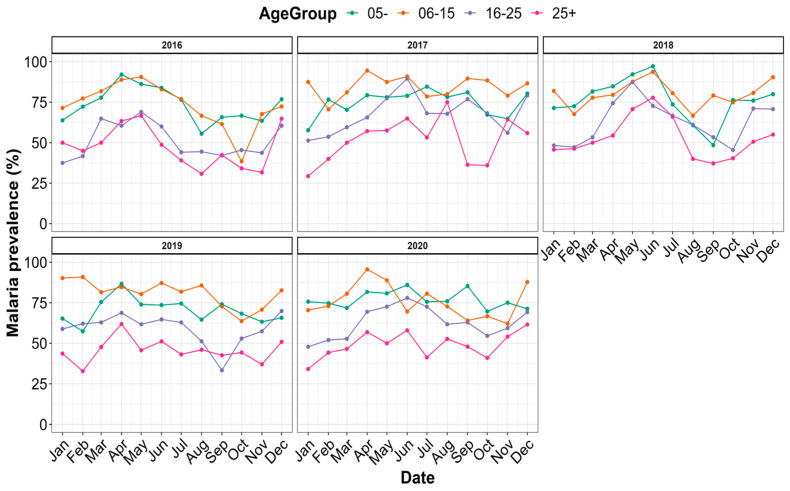
Distribution of malaria prevalence by age strata between 2016 and 2020.

**Table 1 tropicalmed-09-00231-t001:** Distribution of malaria cases by gender and age group over the period of 2016–2020.

Year	Age Group	Gender	
		Female (%)	Male (%)
2016	≤5	233^x^/309^y^ (75.40)	288/377 (76.39)
	6 to 15	149/203 (73.40)	144/176 (81.82)
	16 to 25	111/222 (50.00)	62/105 (59.05)
	>25	234/489 (47.85)	108/231 (46.75)
2017	≤5	247/332 (74.40)	287/384 (74.74)
	6 to 15	166/200 (83)	149/171(87.13)
	16 to 25	183/286 (63.99)	67/93 (72.04)
	>25	278/550 (50.55)	132/250 (52.80)
2018	≤5	271/329 (82.37)	250/332 (75.30)
	6 to 15	220/269 (82.09)	171/215 (79.53)
	16 to 25	187/300 (62.33)	78/114 (68.42)
	> 25	282/524 (53.82)	143/274 (52.19)
2019	≤5	348/481 (72.35)	362/510 (70.98)
	6 to 15	215/268 (80.22)	224/272 (82.35)
	16 to 25	240/403 (59.55)	96/149 (64.43)
	>25	311/669 (46.49)	155/332 (46.69)
2020	≤5	358/470 (76.17)	402/518 (77.61)
	6 to 15	201/256 (78.52)	168/218 (77.06)
	16 to 25	323/523 (61.76)	87/119 (73.11)
	>25	387/758 (51.06)	148/320 (46.25)

^x/y^: x represents the number of malaria-positive cases per age group; y represents the total number of malaria tests performed per age group. Malaria prevalence by Health Center over time (2016–2020).

**Table 2 tropicalmed-09-00231-t002:** Malaria prevalence in 3 health centers of Makenene during the period 2016–2020.

Health Center	Year	Microscopy/RDT *	POS	Prevalence (%)	P/N Ratio
DMC	2016	529	391	73.91	2.83
	2017	1010	672	66.54	1.98
	2018	1278	904	70.73	2.41
	2019	1297	842	64.92	1.86
	2020	1877	1195	63.66	1.75
CHC	2016	1342	769	57.30	1.34
	2017	954	572	59.96	1.5
	2018	854	499	58.43	1.4
	2019	1457	850	58.34	1.4
	2020	1156	773	66.87	2.02
BCHC	2016	432	314	72.70	2.66
	2017	416	339	81.50	4.4
	2018	326	272	83.43	5.04
	2019	349	268	76.79	3.27
	2020	188	135	71.81	2.55

BCHC: Baptist Church Health Center; DMC: District Medical Center; CHC: Catholic Health Center; P/N ratio: number of positive case/negative cases per year after microscopy or RDT POS: positive case; *: number of diagnoses performed. Potential risk factors associated with malaria prevalence.

**Table 3 tropicalmed-09-00231-t003:** Effects of gender and age group on prevalence of exposure to malaria parasite.

Gender	Age Group	Positive (%)	Negative (%)	OR (95% CI)	*p*-Value
Male	≤5	1591 (45.06)	532 (32.34)	1	–
	6 to 15	857 (24.27)	199(12.1)	1.42 (1.19–1.69)	<0.001
	16 to 25	397(11.24)	193 (11.73)	0.62 (0.51–0.84)	<0.001
	> 25	686 (19.43)	721 (43.83)	0.29 (0.27–0.36)	<0.001
	Subtotal	3531 (100)	1645 (100)		
Female	≤5	1457 (29.41)	464 (16.14)	1	–
	6 to 15	951 (19.20)	243(8.45)	1.26 (1.08–1.51)	0.011
	16 to 25	1054 (21.27)	690 (24.01)	0.45 (0.41–0.48)	<0.001
	>25	1492 (30.12)	1477 (51.4)	0.31 (0.31–0.36)	<0.001
	Subtotal	4954 (100)	2874(100)		
Total	≤5	3048 (35.92)	996 (22.04)	1	–
	6 to 15	1808 (21.31)	442 (9.78)	1.32 (1.21–1.55)	<0.001
	16 to 25	1451 (17.10)	883 (19.54)	0.51 (0.42–0.63)	<0.001
	>25	2178 (25.67)	2198 (48.64)	0.29 (0.12–0.39)	<0.001
	Total	8485 (100)	4519(100)		

OR: odds ratio; CI: confidence interval; NB: percentages are calculated based on columns.

**Table 4 tropicalmed-09-00231-t004:** Gender difference in prevalence of exposure to malaria parasite.

Variables	Positive (%)	Negative (%)	OR (95% CI)	*p*
Female	5110 (68.39%)	2983 (31.61%)	1	0.004
Male	3639 (63.14%)	1682 (36.86%)	1.08 (1.02–1.14)	

OR: odds ratio; CI: confidence interval; NB: percentages are calculated based on rows.

## Data Availability

The original contributions presented in the study are included in the article and Appendix A, further inquiries can be directed to the corresponding author.

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
