# Peer review of "A Five-Year Malaria Prevalence/Frequency in Makenene in a Forest–Savannah Transition Ecozone of Central Cameroon: The Results of a Retrospective Study"

_tropicalmed, 2024, doi:10.3390/tropicalmed9100231_

Round 1

Reviewer 1 Report

Comments and Suggestions for Authors

·      The scope of the title appears somewhat narrow. While it mentions a five-year malaria prevalence, the manuscript itself includes not only prevalence data but also an analysis of various other risk factors.

·      Overall, I'm uncertain whether the malaria data in this study should be more appropriately reported as incidence rather than prevalence.

·      Although the authors mention generalized linear models in both the abstract and data analysis sections, the model used seems to resemble a simple logistic regression.

·      The authors discuss about LLIN mass distribution campaigns throughout the manuscript. However, the data and results do not provide sufficient information about these campaigns, nor is there any clear link between LLINs and malaria prevalence in the study.

·      As the study utilized secondary data, did the authors apply any assurance and quality control measures to ensure data quality?

·      Figure 2 should include the actual number of participants.

·      There are two figures labeled as Figure 4 in the manuscript, and the meaning of "Global variation" in both figures is unclear to me.

·      The overall analysis seems somewhat superficial. Additionally, Table 3 lacks clarity—why did the authors choose to compare age groups by gender adjustment only? It would be beneficial to consider other available variables such as gender itself, health facilities, etc. amid limited data availability.

·      Please include a section on the strengths and limitations of the study.

·      The references need to be restructured to align with the journal's in-house style.

Comments on the Quality of English Language

Minor editing of English language required.

Author Response

Reviewer #1 : Commentaries

  • The scope of the title appears somewhat narrow. While it mentions a five-year malaria prevalence, the manuscript itself includes not only prevalence data but also an analysis of various other risk factors.

Answer 1: Thank you for the remark. Although the manuscript analyses risk factors which are very minimal, namely age and sex only; it rather highlights the fact that malaria is stable in the locality with high monthly and annual prevalence over five years. This is the reason that pushed us to choose it as a title. However, we suggest putting the frequency aspect of malaria in the title because the manuscript clearly shows how the frequency of cases (positive and negative) varied over time. Here is a suggestion: “Malaria prevalence/frequency: a result of five years retrospective study in Makenene health area in a forest-savannah transition ecozone, Centre-Cameroun”. Or Trend of malaria at different health centres in Makenene, a forest-savannah transition ecozone of Centre-Cameroun: a retrospective results from routine data from 2016-2020. 

  • Overall, I'm uncertain whether the malaria data in this study should be more appropriately reported as incidence rather than prevalence.

Answer 2: The data reported in this study highlight both prevalence and frequency. This is why, we suggest modifying the title as proposed in the previous comment.

  • Although the authors mention generalized linear models in both the abstract and data analysis sections, the model used seems to resemble a simple logistic regression.

Answer 3: Thank you for the remark. Linear models and logistic regression models were used. Estimation of the risk factors present for example in Table 3 comes from a logistic regression model, while Figures 4 and 5 were generated from a linear model. The statistical analysis paragraph has been rewritten (line 130 to 136), and the correspondence in the abstract (line 19-20). “Graphical analysis was carried out from generalized linear models, to assess the trend of malaria frequency according to categorical variables,  and over time, using the ggplot2 package (Wickham, 2016). The difference in exposure to malaria was estimated through Odds ratios and associated 95 % confidence intervals from the logistic regression model.”

  • The authors discuss about LLIN mass distribution campaigns throughout the manuscript. However, the data and results do not provide sufficient information about these campaigns, nor is there any clear link between LLINs and malaria prevalence in the study.

Answer 4: Thank you for the remark. The study does not clearly show the link between the use of LLINs and the prevalence of malaria, but the LLINs distribution campaign was chosen to justify the study period (2016-2020). Establishing the prevalence of malaria, or showing how prevalence and frequency vary over time in a locality that is not covered under LLIN would raise several other questions/concerns because malaria is endemic in Cameroon. LLINs were distributed in 2016 to the population of Makenene to fight against malaria. Knowing that LLINs have a maximum duration of 3-5 years according to some authors, it was wise to evaluate the frequency of malaria and/or the prevalence over 05 years. The period of 2016-2020 was therefore chosen for this. However, nothing shows that populations regularly used these LLINs, which remains a limitation of this study. A limiting section of the study was thus added to the manuscript as proposed in comment 9 (given that, the data comes from a secondary source).

  • As the study utilized secondary data, did the authors apply any assurance and quality control measures to ensure data quality?

Answer 5: The data used are those transmitted by health centers to the Ministry of Public Health. Obtaining this data is a fairly complex process because it can only be used under the guidance of a specific hierarchical structure. Access to these data was obtained after approval by a regional ethics committee and human health research after validation of the research protocol by a department of the Ministry of Public Health (the regional public health delegation). Ethical clearance No—1289/CRERSHC/2021 is proof of this. The data were faithfully transcribed for use as presented in this manuscript after authorization from the heads of health centres and under the supervision of staff from the hospital. The quality is there. This clarification has been added where appropriate (Data source section).

  • Figure 2 should include the actual number of participants.

Answer 6: Thank you for the remark. The numbers have been added, and the figure has been slightly modified for better understanding as suggested by the Reviewer #2. The section has also been rewritten (see line 130-136).

  • There are two figures labeled as Figure 4 in the manuscript, and the meaning of "Global variation" in both figures is unclear to me.

Answer 7:  Thank you for the remark. The figure labels have been decoupled, and modified. The elements present in Figure 4 which deal with the global variation were introduced in text form, which allows better understanding, and thus increases the clarity of the figures (line 172-178 and 186-189). In addition, these figures (on the global variation of malaria prevalence) are present in additional files.

  • The overall analysis seems somewhat superficial. Additionally, Table 3 lacks clarity—why did the authors choose to compare age groups by gender adjustment only? It would be beneficial to consider other available variables such as gender itself, health facilities, etc. amid limited data availability.

Answer 8:  Thank you for the remark. We did not consider it necessary to compare the prevalence according to gender or even according to health centers since Table 1 already shows an overview of the distribution of prevalence/frequency between gender, and Table 2 shows the distribution according to health centres and even over the time, although the ORs were not calculated in each case. The comparison of the prevalence of malaria according to gender was taken globally and was added (in text form) at the beginning of the paragraph which deals with this part (3.6. Determinants associated with malaria prevalence, line 214-218) for a better understanding the paragraph. “Overall, a significant difference in the general prevalence of malaria was observed between genders. Men are more likely to test positive for malaria than women (OR = 1.08; 95% CI = 1.02-1.14; p-value = 0.004). Moreover, a general prevalence of 68.39% (3639/5321) was observed in men while a general prevalence of 63.14% (5110/8093) was observed in women”.

  • Please include a section on the strengths and limitations of the study.

Answer 9: The section has been added (line 305-312).

  • The references need to be restructured to align with the journal's in-house style.

Answer 10: Done.

  • Minor editing of English language required.

Answer 11:  A review of English language was done.

Reviewer 2 Report

Comments and Suggestions for Authors

The authors analyse the prevalence of malaria in the locality of Makene. The study is done quite well but some things are unclear:

Looking closely at figure 1, checking the numbers well, perhaps 33351 should have been 3351? In addition, the sum of males and females is 13426 (replacing 3351 with 33351) which does not correspond either to the total number of patients (13525), or to the patients whose sex and the result of the diagnostic test are known (13414). I guess he refers to the total number of patients whose sex is known including those whose test result is not known. Given the small difference between the different values, it would be more accurate to indicate in the figure the numbers of patients who are then analyzed in the graphs and tables. In addition, the description of figure 1 should be improved, specifying well what the authors want to highlight.

Redo figure 2, making it easier to understand by indicating the numbers of patients inside each box. If I understand correctly, in tables 1, 2 and 3 only patients whose sex, age and the result of the malaria diagnostic test are known will be used, so in figure 2 the numbers of patients who are eventually analyzed in these tables should be specified. Among other things, as mentioned before, there are very few patients who are excluded. Rewrite paragraph 3.1 more clearly, paying attention to the numbers. I hope I was clear.

Add a figure explaining the climate in the central region of Cameroon, to make it easier to read the article when talking about the dry season and the rainy season (diagram circular). Explaining that the long dry season does not refer to the length of the time but to the intensity of rainfall, unfortunately not everyone is clear.

In line 148 it is written that ‘general malarial prevalence was …’, It is not clear to me what +- 1.84% refers to, it would be useful to understand how it was calculated

Line 153 reverse June and May

Figure 4: Put the legend according to the order of the ‘season colours’ (LDS-SRS-SDS-LRS)

Figure 4: Put the different seasons in temporal order (LDS-SRS-SDS-LRS)

Figure 4: Make a more detailed description of figure  

In the discussion, put the various sections according to the order written in the results, so first the prevalence of malaria according to the seasons, then according to age and finally according to the sex seasons.

Lines 264-265 (The age group of 6-15 years was more affected by malaria followed by the people under 5 years in general) I don't understand from which graph or table this result is derived.

Lines 281-284 : this sentence should be rewritten, this locality' is repeated too many times

Pay attention to the size of the characters in the description of the figures

Add in the abstract what LLINs stand for, even if it is then specified in the introduction.

Author Response

Reviewer #2 : Commentaries

  • Looking closely at figure 1, checking the numbers well, perhaps 33351 should have been 3351? In addition, the sum of males and females is 13426 (replacing 3351 with 33351) which does not correspond either to the total number of patients (13525), or to the patients whose sex and the result of the diagnostic test are known (13414). I guess he refers to the total number of patients whose sex is known including those whose test result is not known. Given the small difference between the different values, it would be more accurate to indicate in the figure the numbers of patients who are then analysed in the graphs and tables. In addition, the description of figure 1 should be improved, specifying well what the authors want to highlight.

Answer 1: Thank you for this remark which escaped us. It is 3351 instead of 3,3351, which makes a total of 13,426 for the sum of patients who consulted a health center for the diagnosis of malaria. Figure 1 just shows the distribution of patients coming to consult a health center in Makenene for cases of malaria, and this is what we would like to highlight initially. Secondly, this figure is also used for the geographical location of the study site (country, region, department, district, health area). The number of patients per section used for the analyzes were instead added in Figure 2 as proposed by the Reviewer #1 and comment 2 below.

  • Redo figure 2, making it easier to understand by indicating the numbers of patients inside each box. If I understand correctly, in tables 1, 2 and 3 only patients whose sex, age and the result of the malaria diagnostic test are known will be used, so in figure 2 the numbers of patients who are eventually analysed in these tables should be specified. Among other things, as mentioned before, there are very few patients who are excluded. Rewrite paragraph 3.1 more clearly, paying attention to the numbers. I hope I was clear.

Answer 2: Figure 2 has been redone as recommended. Indeed, we can now refer to Figure 2 to understand the numbers in Tables 1, 2 and 3. In addition, paragraph 2.1 has been rewritten as proposed by Reviewer #1 to also better understand Figure 2. Paragraph 3.1 has also been rewritten by adding a more detailed description of the data (see line 142-153).

  • Add a figure explaining the climate in the central region of Cameroon, to make it easier to read the article when talking about the dry season and the rainy season (diagram circular). Explaining that the long dry season does not refer to the length of the time but to the intensity of rainfall, unfortunately not everyone is clear.

Answer 3: Thanks for the suggestion. This detail was added in paragraph 2.1 (Study site) which makes it possible to explain that, the distribution of seasons is a function of the intensity of rainfall rather than the duration (in days or months) in the locality.

  • In line 148 it is written that ‘general malarial prevalence was …’, It is not clear to me what +- 1.84% refers to, it would be useful to understand how it was calculated.

Answer 4: Thank you for the remark. The general prevalence was calculated by associating an interval (which was also calculated) to estimate the range within which this prevalence varies over 05 years. However, this was removed and only the annual prevalence’s were retained for a better understanding.

  • Line 153 reverse June and May

Answer 5:  Done. “The highest prevalence peaks were found during the months of June and May (85.42% and 84.74% respectively)”. We proposed this order because the prevalence observed in June is higher than that observed in May, which is justified by the percentages in parentheses. However, the months were reversed in the text (line 158-160).

  • Figure 4: Put the different seasons in temporal order (LDS-SRS-SDS-LRS).

Answer 6:  Thanks for the observation. The legend was made according the to the order of the ‘season colours’ and following the temporal order. The figure labels have been decoupled, and modified. The elements present in Figure 4 which dealt the global variation were introduced in text form, which allows better understanding, and thus increases the clarity of the figures (Reviewer 1).

Figure 4: Make a more detailed description of figure 4 

Answer 7:  A more detailed description of Figure 4 has been made in the appropriate section (3.3. Seasonal Malaria prevalence/frequency, and Variation over Time).

  • In the discussion, put the various sections according to the order written in the results, so first the prevalence of malaria according to the seasons, then according to age and finally according to the sex seasons.

Answer 8:  The order in the discussion has been changed according to the order of the results.

  • Lines 264-265 (The age group of 6-15 years was more affected by malaria followed by the people under 5 years in general) I don't understand from which graph or table this result is derived.

Answer 9:  Thanks for the remark. This sentence has been rewritten. According to Table 3 (last rows), the 6-15-year-old age group was more likely to test positive after a malaria test compared to the age group under 5 in general (OR= 1.32, p<0.005). It was the same when we consider males (OR= 1.42, p<0.005) or females (OR= 1.26, p<0.005) individually; The 6-15 age group was more likely to test positive for malaria than those under 05. In addition, by calculating the prevalence by age groups in General, 80.44%, 74.61%, 61.83%, 49.59% were obtained for the age groups of 6-15, 0-5, 16-25 and > 25 years respectively. This was added in addition to Figure 5 as explained on lines 184 to 186. The figure is also added as an additional file to make the first figure clearer and simpler as proposed by Reviewer #1.

  • Lines 281-284: this sentence should be rewritten, this locality' is repeated too many times.

Answer 10: Done. The sentence has been modified.   

  • Add in the abstract what LLINs stand for, even if it is then specified in the introduction.

Answer 11: Thanks for the remark. It has been added in the abstract (line 17-18).

Round 2

Reviewer 1 Report

Comments and Suggestions for Authors

Thank you to the authors for addressing most of my comments from round 1. However, further revisions are necessary before the manuscript can be accepted for publication. Below are specific areas that need attention:

1.  The authors mentioned using the ggplot2 package for Figures 4 and 5. Please provide more explanation, as I did not see any regression lines or regression values in the figures. Specify the type of logistic regression used for data analysis. If the term "prevalence" is used in the title and throughout the manuscript, please include details on how prevalence was calculated in this study.

2.  Table 3 should include an additional row for negative (%) or column percentages should be described to estimate the OR values accurately.

3.  The current misinterpretation of results affects the discussion. For example, the statement “a higher malaria prevalence per gender was observed in males for younger groups, and in females for the age groups of 6-25 and >25 years” is misleading and does not align with the results from the OR and 95% CI in Table 3. In fact, younger females also had a high risk of malaria infection. The entire paragraph should be rewritten to reflect these findings accurately. The manuscript lacks discussion on the association between health center locations and malaria incidence. This should be addressed.

4. The limitations section should be placed at the end of the discussion without a sub-heading.

5.  If LLIN distribution is to be included, the conclusion should be expanded to summarize significant associations between malaria and factors such as seasonality and patient gender, and suggest targeted interventions for high-risk groups.

6.  All references are still incorrectly formatted, and reference numbers in parentheses in the main text are not properly cited.

Comments on the Quality of English Language

Many language and grammar errors remain. Here are a few visible examples, though numerous other errors exist throughout the manuscript:

Abstract: “Differences in exposure to malaria were estimated through linear and regression logistic models.”

Data Analysis: “All 122 patients with results from malaria diagnostic tests and categorical variables (such as age and sex) were included in the analyzes. Graphical analyzes were performed using generalized linear models to assess the trend of malaria frequency according to categorical variables and over time, using the ggplot2 package. The analyzes were considered significant at the 0.05 level.”

Funding: “This study did not received funding from any sources or agencies.”

Consistency of Terminology: Clarify whether "Cameroun" or "Cameroon" should be used. The title uses "Cameroun."

Table 3 Title: The title of Table 3 regarding global malaria is unclear. Please provide more detail.

Table 2 Decimal Symbols: Verify whether decimal symbols should be commas in Table 2.

Author Response

Manuscript: Trop. Med. Infect. Dis. FOR PEER REVIEW

Title: A five-year malaria prevalence/frequency in Makenene, in a forest-savannah transition ecozone of Centre-Cameroon: a result of a retrospective study

Round #2

Corresponding author: Bamou Roland; bamou2011@gmail.com

Responses to reviewers comment’s

The comments in bold are those of the reviewers present in revised version and the sentences in italics (answers) are those provided by the various authors in response to the questions.

Reviewer #1:

Comment 1: The authors mentioned using the ggplot2 package for Figures 4 and 5. Please provide more explanation, as I did not see any regression lines or regression values in the figures. Specify the type of logistic regression used for data analysis. If the term "prevalence" is used in the title and throughout the manuscript, please include details on how prevalence was calculated in this study.

Answer 1: Thanks for the remark; It was a big mistake in the methodology (statistical analyses). Figures 4 and 5 show (i) the trend of malaria prevalence per month for each year, and (ii) the trend of malaria prevalence per month for each year and age group, respectively. Those two figures (as well as Figure 3) were done with ggplot 2 package and purely descriptive. Therefore, no interferential method (like regression) is applied to them. The statistical analyses paragraph has been rewritten (line 122 to 129).

Data were entered into Excel 2016 spreadsheets, processed, coded and imported to R software (R version 4.2.3, 2023-01-10) for analysis, following the flowchart of figure 2. Patients with the results from malaria diagnostic test and variables such as age and gender, were included in the analyzes. Graphical analyses were performed to assess the trend of malaria prevalence per month for each year, and the trend of malaria prevalence per age group per month for each year respectively, using the ggplot 2 package. Malaria prevalence were calculated by the ratio of positive cases out of the total number per month during each year, and per age group per month per year. Difference in exposure to malaria was estimated through Odds ratios and associated 95 % confidence intervals from logistic regression models. The analyzes were considered significant at the significance level of 0.05.

Comment 2: Table 3 should include an additional row for negative (%) or column percentages should be described to estimate the OR values accurately.

Answer 2: The lines showing negative percentages have been added in the table.

Gender

Age group

Positive (%)

Negative (%)

OR (95% CI)

p-value

Male

≤5

1591 (45.06)

532 (32.34)

1

6 to 15

857 (24.27)

199(12.1)

1.42 (1.19 – 1.69)

<0.001

16 to 25

397(11.24)

193 (11.73)

0.62 (0.51 – 0.84)

<0.001

> 25

686 (19.43)

721 (43.83)

0.29 (0.27 – 0.36)

<0.001

Subtotal

3531 (100)

1645 (100)

Female

≤5

1457 (29.41)

464 (16.14)

1

6 to 15

951 (19.20)

243(8.45)

 1.26 (1.08 – 1.51)

0.011

16 to 25

1054 (21.27)

690 (24.01)

0.45 (0.41 – 0.48)

<0.001

> 25

1492 (30.12)

1477 (51.4)

0.31 (0.31 – 0.36)

<0.001

Subtotal

4954 (100)

2874(100)

Total

≤5

3048 (35.92)

996 (22.04)

1

6 to 15

1808 (21.31)

442 (9.78)

1.32 (1.21 – 1.55)

<0.001

16 to 25

1451 (17.10)

883 (19.54)

0.51 (0.42 – 0.63)

<0.001

> 25

2178 (25.67)

2198 (48.64)

0.29 (0.32 – 0.39)

<0.001

Total

8485 (100)

4519 (100)

Comment 3: The current misinterpretation of results affects the discussion. For example, the statement “a higher malaria prevalence per gender was observed in males for younger groups, and in females for the age groups of 6-25 and >25 years” is misleading and does not align with the results from the OR and 95% CI in Table 3. In fact, younger females also had a high risk of malaria infection. The entire paragraph should be rewritten to reflect these findings accurately. The manuscript lacks discussion on the association between health center locations and malaria incidence. This should be addressed.

Answer 3: Thanks for the remark; it was confusing. The paragraph was rewritten (reflecting the findings accurately) considering the ORs only; because the prevalence were shown in the figures (line 208-213). In addition, Table 4 has been added to present the risk as simply as possible, based on gender only (line 218-222).

 A brief discussion was added to show the association between health centers and malaria prevalence. “It is difficult to compare the results of malaria prevalence according to health centers because of their proximity. However, the high number of patients going to the DMC could be attributed to the fact that, it is the reference health facility in the locality, and the oldest, followed by the CHC. In addition, the prices of care are lower, which sometimes creates a large number of patients during consultations. Therefore, in the event of an emergency, patients tend to go to the least populated health facility to be quickly treated, which could explain the higher prevalence observed at Baptist Church Health Center (BCHC) compared to others health centers”. It has been added in the discussion (line 260- 267).

Comment 4: The limitations section should be placed at the end of the discussion without a sub-heading.

Answer 4: Done. The limitations were placed at the end of the discussion.

Comment 5: If LLIN distribution is to be included, the conclusion should be expanded to summarize significant associations between malaria and factors such as seasonality and patient gender, and suggest targeted interventions for high-risk groups.

Answer 5: Thank you for the remark. The conclusion was expanded by considering the suggested factors (line 298-311).

A high malaria prevalence was observed in Makenene from 2016 to 2020. Although the population of this locality was covered by the second LLIN distribution campaign since 2016, there would be residual malaria transmission capable of maintaining a high level of malaria endemic in this locality. The most visible consequences are observed in children aged 6 to 15 years, then 0 to 5 years, where the highest frequency and prevalence have been observed. The environmental conditions present during the short rainy season could be the origin of the highest prevalence observed during this season compared to other seasons. The probability of being positive for malaria was higher in men as was the overall prevalence. It would be detrimental for public authorities to combine other control methods of malaria with the LLIN in order to reduce the high prevalence observed especially among vulnerable populations such as children and adolescents in Makenene. Additional studies on the bionomics of malaria vectors are necessary to characterize malaria transmission and thus identify the mechanisms that maintain high malaria prevalence in this locality.

Comment 6: All references are still incorrectly formatted, and reference numbers in parentheses in the main text are not properly cited.

Answer 6: The format of references has been changed, both in the main text and in the references section.

Comment 7: Comments on the Quality of English Language. Many language and grammar errors remain.

Answer 7: The quality of English language has been modified throughout the manuscript.

Comment 8: Table 3 Title: The title of Table 3 regarding global malaria is unclear. Please provide more detail.

Answer 8: Tilte has been modified.

Comment 9: Table 2 Decimal Symbols: Verify whether decimal symbols should be commas in Table 2.

Answer 9: Thanks for the remark. It has been changed. 

Reviewer 2 Report

Comments and Suggestions for Authors

Good job! 

Author Response

Thanks 

Round 3

Reviewer 1 Report

Comments and Suggestions for Authors

1. I found several typos or language errors, such as:
- In the abstract: "regression logistics"
- In the data analysis section: "analyzes"
- In the funding section: "did not received"

  2. Some references are still incorrectly formatted (e.g., journal names).

3. In Table 3, one OR value or its 95% confidence interval is incorrect: [0.29 (0.32-0.39)].

Comments on the Quality of English Language

I found several typos or language errors, such as:
- In the abstract: "regression logistics"
- In the data analysis section: "analyzes"
- In the funding section: "did not received"

Author Response

Round #3

Corresponding author: Bamou Roland; bamou2011@gmail.com

Responses to reviewers ’ comments

The comments in bold are those of the reviewers present in the revised version and the sentences in italics (answers) are those provided by the various authors in response to the questions.

Comment 1. I found several typos or language errors, such as:

- In the abstract: "regression logistics"

- In the data analysis section: "analyzes"

- In the funding section: "did not receive"

Answer 1: Thank you for the remark. Language errors have been corrected in the appropriate sections (Abstract, data analysis and funding)

Comment 2. Some references are still incorrectly formatted (e.g., journal names).

Answer 2: Thank you for the remark. The references were formatted according to the journal form (Trop. Med. Infect. Dis.).

Comment 3. In Table 3, one OR value or its 95% confidence interval is incorrect: [0.29 (0.32-0.39)].

Answer 3: Thanks. It was a big typo. The confidence interval is (0.12 -0.39) not (0.32- 0.29). It has been changed in the Table.